METHODS

# Integrative Mendelian randomization for detecting exposure-by-group interactions using group-specific and combined summary statistics

Ke Xu[1,2], Nathaniel Maydanchik[1], Bowei Kang[1], Jianhai Chen[1], Qixiang Chen[1], Gongyao Xu[1], Shinya Tasaki[3], David A. Bennett[3], Lin S. Chen[1]*

1 Department of Public Health Sciences, The University of Chicago, Chicago, Illinois, United States of America, 2 Department of Applied and Computational Mathematics and Statistics, University of Notre Dame, Indiana, United States of America, 3 Department of Neurological Sciences and Rush Alzheimer's Disease Center, Rush University Medical Center, Chicago, Illinois, United States of America

* lchen4@bsd.uchicago.edu

**Data availability statement:** All the GWAS summary statistics of IV-to-exposure effects

## Abstract

Interactions between risk factors and covariate-defined groups are commonly observed in complex diseases. Existing methods for detecting interactions typically require individual-level data. The data availability and the measurements of risk exposures and covariates often limit the power and applicability in assessing interactions. To address these limitations, we propose int2MR, an integrative Mendelian randomization (MR) method that leverages GWAS summary statistics on exposure traits and group-separated and/or combined GWAS statistics on outcome traits. The int2MR can assess a broad range of risk exposure effects on diseases and traits, revealing interactions unattainable with incomplete or limited individual-level data. Simulation studies demonstrate that int2MR effectively controls type I error rates under various settings while achieving considerable power gains with the integration of additional group-combined GWAS data. We applied int2MR to two data analyses. First, we identified risk exposures with sex-interaction effects on ADHD, and our results suggested potentially elevated inflammation in males. Second, we detected age-group-specific risk factors for Alzheimer's disease pathologies in the oldest-old (age 95+); many of these factors were related to immune and inflammatory processes. Our findings suggest that reduced chronic inflammation may underlie the distinct pathological mechanisms observed in this age group. The int2MR is a robust and flexible tool for assessing group-specific or interaction effects, providing insights into disease mechanisms.

## Author summary

Complex diseases often arise from interactions between genetic, environmental, and biological factors, leading to differential effects across population subgroups. However, existing methods for detecting such interactions typically require access to

used in this paper are publicly available. Related links to the summary statistics can be found in S1 Table. The R implementation of our int2MR method is available at https://github.com/kxu-stat/int2MR. Our implementation of algorithms depends on `rstan` (available on https://CRAN.R-project.org/package=rstan). Additionally, the GWAS summary statistics for the IV-to-outcome effects derived from the Religious Orders Study and the Rush Memory and Aging Project (ROSMAP), along with all primary results underlying our analyses, have been deposited on Zenodo at https://doi.org/10.5281/zenodo.16341091.

**Funding:** This work was supported by the National Institutes of Health (grant 1R01GM154421 to LSC, BK, JC, and QC; grant 1U01MH139345 to LSC, BK, JC, and QC). The funders had no role in study design, data collection and analysis, decision to publish, or preparation of the manuscript.

**Competing interests:** The authors have declared that no competing interests exist.

individual-level data, which is often limited or incomplete. To address this challenge, we propose int2MR, an integrative Mendelian Randomization (MR) method that leverages GWAS summary statistics from both exposure traits and outcome traits, including group-separated and combined GWAS data. int2MR enables the detection of exposure-by-group interaction effects even when individual-level data are unavailable. Through simulation studies, we demonstrate that int2MR effectively controls type I error rates while achieving substantial power gains when integrating group-combined GWAS data. We apply int2MR in two real-world analyses: (1) identifying risk exposures with sex-differentiated effects on ADHD and (2) detecting age-differentiated risk factors for Alzheimer's disease. Our findings highlight the utility of int2MR in uncovering previously undetectable interactions, improving statistical power, and enhancing the interpretability of genetic association studies for complex diseases.

## Introduction

Complex diseases often arise from a combination of genetic, environmental, and biological factors, resulting in varied effects of risk factors across different subgroups [1,2]. These group-specific risk effects, also known as exposure-by-group interaction effects, play a critical role in disease mechanisms [3,4]. For example, genetic variations may influence susceptibility to risk factors such as diet, pollution, or lifestyle within specific populations [5–11]. Biological differences, such as age and sex, may lead to different disease pathways across groups [12–21]. Additionally, social determinants of health, including access to healthcare and socioeconomic status, may shape exposure-disease relationships differently for various groups [22–24]. Understanding these exposure-by-group interaction effects is crucial for advancing research on disease mechanisms and precision medicine. By considering group-specific effects, precision medicine can better address the unique risks of diverse populations and improve health outcomes, especially for vulnerable populations.

To assess exposure-by-group interaction effects, a common approach is interaction analysis, which estimates both the effects of risk exposures and their interactions with groups or covariates on disease outcomes [25]. These analyses, though, are often constrained by small sample sizes and the limited availability of risk exposure and covariate measurements within individual studies. Moreover, unmeasured confounding variables can bias the results, complicating causal interpretations. Two-stage least squares (2SLS) methods have been applied to detect interactions while allowing for unmeasured confounding under certain assumptions [26]. However, their applicability is also restricted to studies with individual-level data, and the power is constrained by the strength of the instruments in the data. Mendelian randomization (MR) is a powerful tool to evaluate the causal effects of risk exposures on complex disease outcomes, treating genetic variants associated with exposures of interest as instrumental variables (IVs) [27–30]. Two-sample MR, which uses two sets of genome-wide association study (GWAS) summary statistics as input, has achieved many successes in assessing the causal effects of complex traits as exposures on various diseases as outcomes [31–42]. Most existing MR methods are designed to assess total effects, with limited focus on detecting exposure-by-group interaction effects. While some recent MR methods were proposed to detect interaction effects [3,4], they require individual-level data on exposure, outcome, and the covariate group variables. The growing availability of GWAS summary statistics highlights the need for two-sample MR methods to detect interactions across covariate-defined groups using summary data.

Here we propose an *int*egrative MR method for detecting *int*eraction effects (int2MR) between exposures and covariate groups on complex diseases, using only summary statistics as input. The int2MR method integrates group-specific and group-combined GWAS summary statistics from multiple studies and consortia, enhancing the power to detect group-specific effects of risk exposures. Through extensive simulations and method comparisons, we demonstrated the advantages of the proposed int2MR method for detecting interactions and main effects. We applied the proposed int2MR method to two data analyses. In the first analysis, by integrating sex-stratified and sex-combined Attention Deficit Hyperactivity Disorder (ADHD) GWAS summary statistics from the Psychiatric Genomics Consortium [43,44] and other major consortia, we boosted the power for identifying exposures with sex-interaction effects on ADHD. In the second analysis, motivated by the observation that AD pathology peaks around age 95 and then declines, we sought to identify risk factors with age-group-specific effects on AD pathology in the oldest-old (defined as death-age 95+) compared to the rest of the population. Using int2MR, we integrated age-group-stratified GWAS summary statistics from ROSMAP (Religious Orders Study and the Rush Memory and Aging Project) [45] with publicly available GWAS summary statistics on a wide range of risk exposures from major consortia. We identified multiple immune and inflammation-related exposures with age group-differential effects on AD pathology in the oldest-old, suggesting reduced inflammation and potentially distinct neuroinflammatory mechanisms in the oldest-old compared with the rest of the population. These analyses demonstrate int2MR's flexibility in assessing exposure-by-group interactions across diverse traits and groups (e.g., socioeconomic or environmental factors) using GWAS summary statistics. It enhances detection power for main and interaction effects by integrating group-stratified and group-combined GWAS data.

## Description of the method

We propose the int2MR model to detect exposure-by-group interaction effects. We introduce two variations of the model, as illustrated in Fig 1. In this section, we will describe the input statistics and then present the model formulation and the estimation algorithms for both variations. Furthermore, the model is flexible and can be extended to integrate IV-to-outcome statistics from mixed groups with varying group compositions (see S1 Text for details).

### int2MR: An integrative MR for detecting exposure-by-group interaction effects with group-separated GWAS summary statistics

As illustrated in Fig 1A, the first variation of int2MR takes the following summary statistics as input:

- IV-to-exposure statistics: Standard GWAS summary statistics for the exposure trait.
- Group-specific IV-to-outcome statistics: GWAS summary statistics for the outcome stratified by groups (i.e. reference group and comparison group).

For the $j$-th SNP ($j = 1, 2, \cdots, p$), let $\left\{ \widehat{\gamma}_j, \widehat{s}_{\gamma,j} \right\}_{j=1}^{p}$ denote the observed marginal IV-to-exposure effect and its standard error obtained from a GWAS study for the exposure trait. Similarly, let $\left\{ \widehat{\Gamma}_{0,j}, \widehat{s}_{0,j} \right\}_{j=1}^{p}$ and $\left\{ \widehat{\Gamma}_{1,j}, \widehat{s}_{1,j} \right\}_{j=1}^{p}$ denote the observed marginal IV-to-outcome effects and their standard errors for the reference group (denoted by subscript 0) and the comparison group (denoted by subscript 1), respectively. The IV-to-outcome statistics for the two groups are obtained from group-stratified GWAS studies on the outcome trait. Suppose that $\left\{ \gamma_j \right\}_{j=1}^{p}$ are the true IV-to-exposure effects, and $\left\{ \Gamma_{0,j} \right\}_{j=1}^{p}$ and $\left\{ \Gamma_{1,j} \right\}_{j=1}^{p}$ are the true IV-to-outcome effects for the reference and comparison groups, respectively. For the $j$-th SNP as an

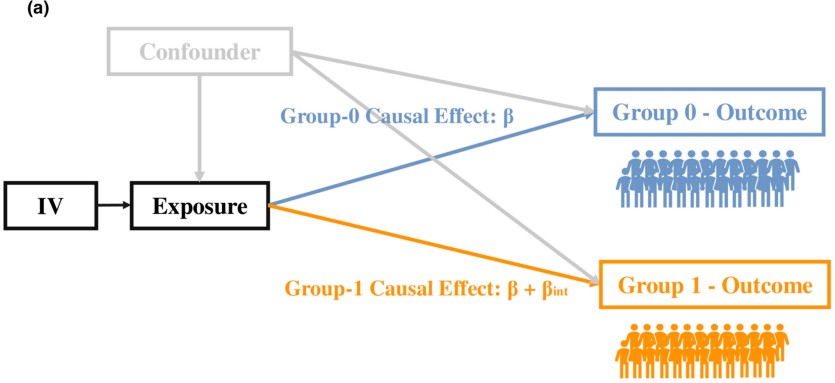

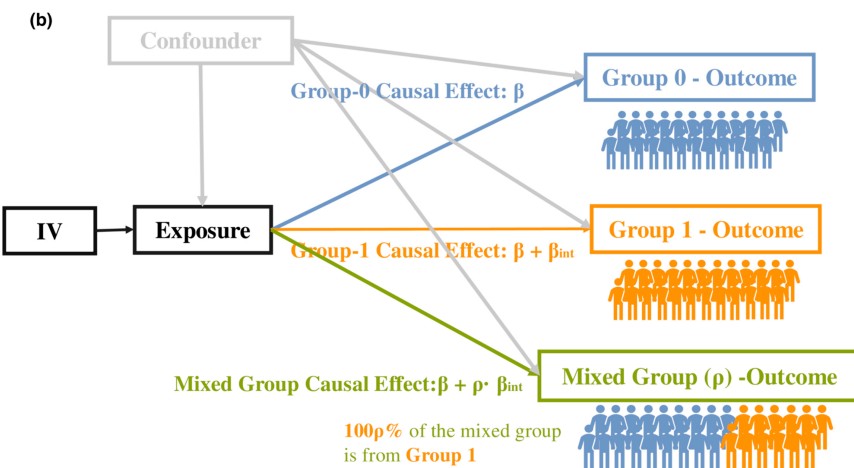

**Fig 1. Illustrations of the proposed int2MR framework for detecting interaction effects.** (a) The causal diagram of the int2MR model, using summary statistics from two group-specific IV-to-outcome GWASs as input. The model estimates the causal effect $\beta$ for the reference group (Group 0) and $\beta + \beta_{int}$ for the comparison group (Group 1), capturing group-specific effects. (b) int2MR using two sets of group-specific IV-to-outcome GWAS statistics and one set of group-combined GWAS statistics for input. The parameter $\rho$ represents the proportion of samples from the comparison group in the mixed-group study. The IV-to-exposure GWAS statistics in int2MR analyses are all from standard group-combined GWAS analyses.

IV, we jointly model the IV-to-exposure effect and the IV-to-outcome effects in the reference and comparison groups as:

$$\begin{pmatrix} \widehat{\gamma}_j \\ \widehat{\Gamma}_{0,j} \\ \widehat{\Gamma}_{1,j} \end{pmatrix} \sim \mathcal{N}\left( \begin{pmatrix} \gamma_j \\ \Gamma_{0,j} \\ \Gamma_{1,j} \end{pmatrix}, \begin{pmatrix} \widehat{s}_{\gamma,j}^2 & & \\ & \widehat{s}_{0,j}^2 & \\ & & \widehat{s}_{1,j}^2 \end{pmatrix} \right). \tag{1}$$

Our framework assumes that IVs must satisfy the standard core assumptions: (1) Relevance: the IVs are associated with the exposure (i.e., the IV-to-exposure effects, $\gamma_j$, are nonzero and assumed homogeneous across groups), (2) Independence: the IVs are independent of any unmeasured confounders, and (3) Exclusion Restriction: the IVs affect the outcome only through their effects on the exposure. We relax this last assumption by allowing

for uncorrelated pleiotropy. We assume balanced pleiotropy; that is, any residual direct effects of the IV on the outcome are assumed to average out. In our model, a "valid" IV is one that meets these conditions.

The current model assumptions indeed imply that the group-specific genetic effect on the outcome is fully mediated by the effect of the exposure on the outcome. There is no additional direct group-specific genetic effect on the outcome. In other words, our current model does not allow any additional group-specific genetic effect on the outcome beyond the causal effect $\beta$, and the differential effect across groups is entirely captured through the interaction effect $\beta_{\text{int}}$.

The true IV-to-outcome effects in the two groups are:

$$\text{Reference group (Group 0)}: \quad \Gamma_j = \beta \cdot \gamma_j + \alpha_{0,j},$$
$$\text{Comparison group (Group 1)}: \quad \Gamma_j = (\beta + \beta_{\text{int}}) \cdot \gamma_j + \alpha_{1,j}.$$

In our model, we assume relevance and independence of the IVs, while relaxing the exclusion restriction by allowing for uncorrelated pleiotropy [32,37]. Here, $\alpha_{0,j} \sim N(0, \sigma_{\alpha,0}^2)$ and $\alpha_{1,j} \sim N(0, \sigma_{\alpha,1}^2)$ represent uncorrelated pleiotropic effects in the reference group and comparison group, respectively. The main effect $\beta$ captures the causal effect in the reference group, while $\beta_{\text{int}}$ represents the differential effect between the comparison and reference groups, i.e., the interaction effects. This framework allows for estimating and testing any linear combination of $\beta$ and $\beta_{\text{int}}$. For example, we can separately test for non-zero group-specific causal effects, $\beta$ and $\beta + \beta_{\text{int}}$, for the reference and comparison groups, respectively. Additionally, we can test for non-zero interaction effects, $\beta_{\text{int}}$. Note that the off-diagonal elements in the covariance matrices of Eq (1) are non-zero if there are sample overlaps among different GWAS studies.

## Modeling interaction effects in the group-combined GWAS: from individual-level data to GWAS summary statistics

In the previous section, we focus on the scenario in which group-specific IV-to-outcome statistics are available. In practice, however, a GWAS cohort often includes individuals from both groups, yielding only combined summary statistics. This situation therefore requires us to model interaction effects directly in the group-combined GWAS.

Let $X$ represent the risk factor of interest and $Y$ denote the outcome. At the individual level, we have the following structural equations.

$$X \mid G, S, U = \sum_{j=1}^{p} \gamma_j G_j + \beta_{U_X} \cdot U + \varepsilon_X \tag{2}$$

$$Y \mid X, S, G, U = \beta X + \sum_{j=1}^{p} \alpha_j G_j + \beta_{U_Y} U + \beta_{SY} \cdot S + \beta_{\text{int}} \cdot X \circ S + \varepsilon_Y. \tag{3}$$

In Eqs 2 and 3, the vector $G = (G_1, G_2, \ldots, G_p)$ contains $p$ genetic variants for an individual, and $a = (\alpha_1, \ldots, \alpha_p)$ represents the uncorrelated pleiotropy effects. Here, $S$ denotes a binary group label such as sex, taking values in $\{0, 1\}$, and $U$ is an unmeasured confounder. $\beta_{U_X}$ and $\beta_{U_Y}$ represent the effects of the confounder $U$ on $X$ and $Y$, respectively, and $\beta_{SY}$ represents the direct effect of group $S$ on the outcome (capturing baseline group differences). The parameters of interest, $\beta$ and $\beta_{\text{int}}$, capture the main causal effect of exposure $X$ on outcome $Y$ and the interaction effect between $X$ and group label $S$, respectively.

When only summary statistics are available, individual group labels $S$ are unobserved. Instead, let $\rho$ denote the proportion of the comparison group in the GWAS sample. For instance, if we aim to estimate the female-specific effect $\beta$ in a sex-combined GWAS, then $\rho$ is the proportion of males. We model $S \sim \text{Bernoulli}(\rho)$     independently of $G$ and $U$, so that $\rho = \mathbb{E}[S]$. Here $\rho$ serves as a summary statistic for the unobserved $S$ and can be used to model interaction effects. Substituting Eq 2 into Eq 3 and marginalizing over $S$ yields (see S1 Text for details):

$$Y \mid G, U = \widetilde{c}_0 + \sum_{j=1}^{p} \left[ (\beta + \beta_{\text{int}} \cdot \rho) \, \gamma_j + \alpha_j \right] \cdot G_j + \widetilde{c}_1 \cdot U + \widetilde{\varepsilon}_Y, \tag{4}$$

where $\widetilde{c}_0$ is a constant depending on the group indicator $S$, $\widetilde{c}_1$ is a constant representing the strength of the confounding effect. Here, $\widetilde{c}_0$ and $\widetilde{c}_1$ represent the non-genetic contributions.

Here, we denote the true IV-to-outcome effect $\Gamma_j$ for the $j$-th SNP, which is defined as the coefficient of $G_j$ in the model Eq 4, i.e.,

$$\Gamma_j = (\beta + \beta_{\text{int}} \cdot \rho) \, \gamma_j + \alpha_j \tag{5}$$

for any $j = 1, 2, \cdots, p$, where $\gamma_j$ is the $j$-th IV-to-exposure effect. We denote the estimated IV-to-outcome and IV-to-exposure effects as $\widehat{\Gamma}_j$ and $\widehat{\gamma}_j$, respectively.

Crucially, each SNP $G_j$ has coefficient $(\beta + \beta_{\text{int}} \cdot \rho) \, \gamma_j + \alpha_j$. Thus, GWAS summary statistics encode the interaction effect $\beta_{\text{int}}$ only through the shift $\rho \cdot \beta_{\text{int}}$ in the effective exposure coefficient. This observation motivates our flexible strategy to recover both $\beta$ and $\beta_{\text{int}}$ by integrating information from combined or group-specific GWAS results.

## Flexible int2MR allowing the integration of group-specific and group-combined GWAS statistics

The proposed int2MR model can integrate group-specific and group-combined GWAS statistics for the outcome disease/trait, enhancing the power and flexibility of the analysis. As illustrated in Fig 1B, the second variation of int2MR can utilize the following types of summary statistics as inputs:

- IV-to-exposure statistics: Standard GWAS summary statistics for the exposure trait.
- Group-specific IV-to-outcome statistics: GWAS summary statistics for the outcome stratified by group, which could be statistics specific to the reference group, the comparison group, or both.
- Group-combined IV-to-outcome statistics: Standard GWAS summary statistics for the outcome trait derived from mixed-group samples, with a known proportion of group composition, $\rho$.

For the $j$-th SNP as an IV, we jointly model the IV-to-outcome effect for each input study and the IV-to-exposure effect as follows:

$$\begin{pmatrix} \widehat{\gamma}_j \\ \widehat{\Gamma}_{0,j} \\ \widehat{\Gamma}_{1,j} \\ \widehat{\Gamma}_{2,j} \end{pmatrix} \sim \mathcal{N} \left( \begin{pmatrix} \gamma_j \\ \Gamma_{0,j} \\ \Gamma_{1,j} \\ \Gamma_{2,j} \end{pmatrix}, \begin{pmatrix} \widehat{s}_{\gamma_j}^2 & & & \\ & \widehat{s}_{0,j}^2 & & \\ & & \widehat{s}_{1,j}^2 & \\ & & & \widehat{s}_{2,j}^2 \end{pmatrix} \right). \tag{6}$$

Note that the covariance matrices above may have non-zero off-diagonal elements if the input GWAS datasets have overlapping samples. These details were omitted here for clarity in presenting the main model.

The true IV-to-outcome effect in each study with varying proportions of comparison group samples is specified as:

$$\Gamma_{k,j} = \left( \beta + \beta_{\text{int}} \cdot \rho_k \right) \gamma_j + \alpha_{k,j} \quad \text{for} \quad k \in \{0, 1, 2\},$$

where $\rho_k$ represents the proportion of comparison group samples in the $k$-th study. For the study with only reference group samples ($k = 0$), $\rho_0 = 0$; for the study with only comparison group samples ($k = 1$), $\rho_1 = 1$; and for the study with group-combined samples ($k = 2$), $\rho_2$ reflects the proportion of the comparison group samples. Here, $\alpha_{k,j} \sim N(0, \sigma_{\alpha,k}^2)$ is the uncorrelated pleiotropic effect for each study. The causal effect of the reference group is $\beta$ and the causal effect of the comparison group is $\beta + \beta_{\text{int}}$. The interaction effect $\beta_{\text{int}}$ captures the difference in causal effects comparing the comparison group versus the reference group.

To obtain the parameter estimates and inference of the above models, we build a Bayesian hierarchical model. We implement a No-U-Turn sampler (NUTS) [46,89], a variant of the Hamiltonian Monte Carlo method [47], to generate posterior samples for inference. By the Bernstein–von Mises theorem, the posterior distribution asymptotically approaches a normal distribution. We estimate the standard errors of $\beta$ and $\beta_{\text{int}}$ by inverting the observed Fisher information matrix, denoted as $\mathbf{I}\left(\widehat{\Theta}\right)$, where $\Theta = (\beta, \beta_{\text{int}}, \gamma, \alpha_0, \alpha_1, \alpha_2)$ represents the vector of all model parameters. The observed Fisher information matrix is computed from the Hessian of the negative posterior log-likelihood, yielding $\text{SE}\left(\beta\right) = \sqrt{\mathbf{I}\left(\widehat{\Theta}\right)_{1,1}^{-1}}$ and $\text{SE}\left(\beta_{\text{int}}\right) = \sqrt{\mathbf{I}\left(\widehat{\Theta}\right)_{2,2}^{-1}}$. Further algorithmic details are provided in the S1 Text.

The model described by Eq (1) integrates group-separated GWAS statistics on outcome and the model described by Eq (6) integrates group-separated and combined GWAS statistics on outcome. Additionally, int2MR is generalized to integrate group-combined GWAS statistics on outcomes from two or more studies with varying group compositions. $\beta$ and $\beta_{\text{int}}$ are identifiable as long as the $\rho$ values differ among the IV-to-outcome GWAS samples (see S1 Text for more details). For instance, when analyzing causal effects of male and female groups, we can integrate GWAS statistics on outcome from the Million Veteran Program (MVP) [48,49], which consists of approximately 91.8% males, with another GWAS with an equal proportion of males and females. This flexibility allows int2MR to efficiently combine data from diverse sources, enhancing statistical power and enabling robust inference.

Compared to existing MR methods for detecting interaction effects [3,4], a key innovation and major advantage of our model is that it requires only summary statistics. This approach provides unparalleled flexibility in assessing group-specific effects of risk factors on complex diseases, even when individual-level data on risk exposures are incomplete or unavailable. For example, in our data analysis, we evaluated age-group-specific risk factors for Alzheimer's disease (AD), with a focus on the oldest-old (death-age 95+). Using the ROSMAP study, we obtained age-group-separated GWAS statistics on AD for the oldest-old and those for all samples in ROSMAP as group-specific and combined IV-to-outcome statistics, respectively. Publicly available GWAS statistics for various complex traits, sourced from other GWAS consortia, served as IV-to-exposure statistics. Notably, many of these risk exposures are not directly measured in the ROSMAP data. Yet, our method can assess their interaction effects and age-group-specific contributions to AD and pathologies.

## Allowing correlated SNPs as IVs by accounting for linkage disequilibrium (LD)

We extend the int2MR model to allow moderately correlated SNPs as IVs by modeling the LD structure:

$$\widehat{\gamma} \sim \mathcal{N}\left(\widehat{S}_\gamma \widehat{R} \widehat{S}_\gamma^{-1} \gamma, \widehat{S}_\gamma \widehat{R}_\gamma\right),$$
$$\widehat{\Gamma}_k \sim \mathcal{N}\left(\widehat{S}_{\Gamma_0} \widehat{R}_{\Gamma_k} \widehat{S}_{\Gamma_0}^{-1} \Gamma_k, \widehat{S}_{\Gamma_k} \widehat{R}_{\Gamma_k}\right), k = 0, 1, 2.$$

where $\widehat{\gamma} = \left(\widehat{\gamma}_1, \widehat{\gamma}_2, \cdots, \widehat{\gamma}_p\right)^\top$ and $\gamma = \left(\gamma_1, \gamma_2, \cdots, \gamma_p\right)^\top$ are the vectors of estimated and true marginal IV-to-exposure effects, respectively. $\widehat{\Gamma}_k = \left(\widehat{\Gamma}_{k,1}, \widehat{\Gamma}_{k,2}, \cdots, \widehat{\Gamma}_{k,p}\right)^\top$ and $\Gamma_k = \left(\Gamma_{k,1}, \Gamma_{k,2}, \cdots, \Gamma_{k,p}\right)^\top$ are the vectors of estimated and true marginal IV-to-outcome effects in the $k$-th GWAS sample ($k \in \{0, 1, 2\}$). $\widehat{S}_\gamma, \widehat{S}_{\Gamma_0}, \widehat{S}_{\Gamma_1}$ and $\widehat{S}_{\Gamma_2}$ are the corresponding diagonal matrices of standard errors; and $\widehat{R}_\gamma, \widehat{R}_{\Gamma_0}, \widehat{R}_{\Gamma_1}$ and $\widehat{R}_{\Gamma_2}$ are the corresponding estimated correlation matrices among all selected IVs, which could be estimated using an independent reference panel data. This framework allows for weak to moderate correlations among IVs by accounting for their correlation structures. However, the application of int2MR may not be suited when IVs are highly correlated.

## Verification and comparison

We evaluated the performance of our proposed summary-statistics-based MR method, int2MR, by comparing it with existing methods that utilize either individual-level data or summary statistics. We simulated individual-level genotype data for GWASs of both the exposure and the outcome traits. For the outcome GWAS, we simulated GWAS datasets with group-specific effects, and generated both group-specific and group-combined GWAS summary statistics. In each simulation, we generated 300 SNPs as IVs. For the group-specific GWAS, we simulated 1,000 individuals for the comparison group and 2,000 for the reference group, and each group has group-specific effects. We calculated group-combined GWAS summary statistics by combining data from both the reference group and the comparison group. The total sample size for the mixed group was varied to investigate its effects on method performance, in particular power. In the group-combined GWAS data, individuals from the reference and comparison groups were represented in equal proportions.

From the simulated datasets, we obtained the IV-to-exposure GWAS summary statistics, as well as group-separated and group-combined GWAS summary statistics for the outcome. These summary statistics were used as inputs for our int2MR analyses. We selected the appropriate input data type, either individual-level data or summary statistics, for specific comparison method (see S1 Text for additional simulation details). We evaluated the type I error rates and power using a $p$-value threshold of 0.05.

### Type I error

Table 1A compares the type I error rates of int2MR using group-separated GWASs with int2MR$_{+20k}$ integrating an additional group-combined GWAS consisting of 20,000 mixed-group samples. Additionally, we compared with the 2SLS method for interactions[26] and standard interaction tests based on ordinary least squares (OLS) regression. We simulated various settings with and without horizontal pleiotropy and with varying strengths of confounding effects. In all settings, the proposed int2MR approach, using both group-separated

**Table 1. Simulation results comparing the type I error rates of different methods in different settings: with and without horizontal pleiotropy effects ($\alpha$) and in the presence of confounders ($U$).**

| Method | Testing $H_0 : \beta = 0$ | | | Testing $H_0 : \beta_{int} = 0$ | | |
|---|---|---|---|---|---|---|
| | $\alpha = 0$, $U_0 = 0$, $U_1 = 0.2$ | $\alpha = 0.02$, $U_0 = 0$, $U_1 = 0.2$ | $\alpha = 0.02$, $U_0 = 0.1$, $U_1 = -0.2$ | $\alpha = 0$, $U_0 = 0$, $U_1 = 0.2$ | $\alpha = 0.02$, $U_0 = 0$, $U_1 = 0.2$ | $\alpha = 0.02$, $U_0 = 0.1$, $U_1 = -0.2$ |
| int2MR$_{+20k}$ | 0.048 | 0.060 | 0.066 | 0.048 | 0.062 | 0.066 |
| int2MR | 0.060 | 0.060 | 0.067 | 0.046 | 0.064 | 0.062 |
| 2SLS | 0.054 | 0.078 | 0.100 | 0.030 | 0.082 | 0.078 |
| OLS | 0.072 | 0.064 | 0.082 | 0.132 | 0.148 | 0.044 |
| **Method** | $\alpha_0 = 0$, $\alpha_1 = 0$, $U = 0.2$ | $\alpha_0 = 0$, $\alpha_1 = 0.02$, $U = 0.2$ | $\alpha_0 = 0.02$, $\alpha_1 = 0.02$, $U = 0.2$ | $\alpha_0 = 0$, $\alpha_1 = 0$, $U = 0.2$ | $\alpha_0 = 0$, $\alpha_1 = 0.02$, $U = 0.2$ | $\alpha_0 = 0.02$, $\alpha_1 = 0.02$, $U = 0.2$ |
| int2MR$_{+20k}$ | 0.042 | 0.040 | 0.068 | 0.034 | 0.066 | 0.070 |
| int2MR | 0.042 | 0.040 | 0.062 | 0.046 | 0.064 | 0.072 |
| 2SLS | 0.072 | 0.066 | 0.082 | 0.042 | 0.080 | 0.088 |
| OLS | 0.260 | 0.258 | 0.210 | 0.098 | 0.104 | 0.130 |

**(a)** Type I error rate comparisons for testing interaction effects ($\beta_{int}$), in the presence of (group-specific) confounding and pleiotropy.

**Testing Total Effects ($H_0 : \beta + \beta_{int} = 0$)**

| Method | $\alpha = 0$, $U_0 = 0$, $U_1 = 0.2$ | $\alpha = 0.02$, $U_0 = 0$, $U_1 = 0.2$ | $\alpha = 0.02$, $U_0 = 0.2$, $U_1 = -0.2$ | $\alpha_0 = 0$, $\alpha_1 = 0$, $U = 0.2$ | $\alpha_0 = 0$, $\alpha_1 = 0.02$, $U = 0.2$ | $\alpha_0 = 0.02$, $\alpha_1 = 0.02$, $U = 0.2$ |
|---|---|---|---|---|---|---|
| int2MR$_{+20k}$ | 0.058 | 0.063 | 0.060 | 0.036 | 0.060 | 0.064 |
| int2MR | 0.048 | 0.057 | 0.062 | 0.058 | 0.053 | 0.062 |
| IVW | 0.058 | 0.072 | 0.068 | 0.054 | 0.066 | 0.084 |
| MR-Egger | 0.040 | 0.066 | 0.040 | 0.050 | 0.050 | 0.042 |
| MR-Median | 0.032 | 0.022 | 0.030 | 0.022 | 0.028 | 0.046 |
| MR-RAPS | 0.048 | 0.072 | 0.070 | 0.054 | 0.066 | 0.084 |
| MR-cML | 0.060 | 0.076 | 0.070 | 0.056 | 0.062 | 0.084 |

**(b)** Type I error rate comparisons for testing total effects, in the presence of (group-specific) confounding and pleiotropy.

For int2MR, we integrated group-separated GWASs for reference and comparison groups on outcome. In int2MR$_{+N}$, the subscript "+N" represents the additional sample size from the group-combined GWAS. Among the competing methods, individual-level data-based approaches, such as 2SLS and OLS-based interaction tests, can test for interaction effects, while summary-statistics-based MR methods are limited to testing total effects.

and group-combined GWAS statistics for outcomes, controlled type I error rates. In contrast, the 2SLS method for interactions failed to control type I error rates in the presence of pleiotropy, and the OLS-based interaction test failed to control type I error rates in the presence of confounding. Table 1B compares type I error rates for testing non-zero total effects for int2MR and other existing MR methods under group-specific pleiotropy and confounding. For int2MR and int2MR$_{+20k}$, the null hypothesis $H_0 : \beta = 0$ was tested. For competing MR methods, IVW [31], MR-Egger [32], MR-Median [50], MR-RAPS [33], and MR-cML [36], we tested for total effects since these methods do not assess interactions. The proposed int2MR methods consistently controlled type I error rates, whereas some competing methods showed slight inflation in type I error rates under group-specific pleiotropy or confounding.

## Power comparison

Fig 2A compares the power of different methods for detecting interaction effects. While 2SLS and OLS require individual-level data, int2MR demonstrated comparable power to the OLS-based interaction test with 2,000 samples from the reference group and 1,000 from the comparison group. When additional group-combined GWAS statistics with larger sample sizes

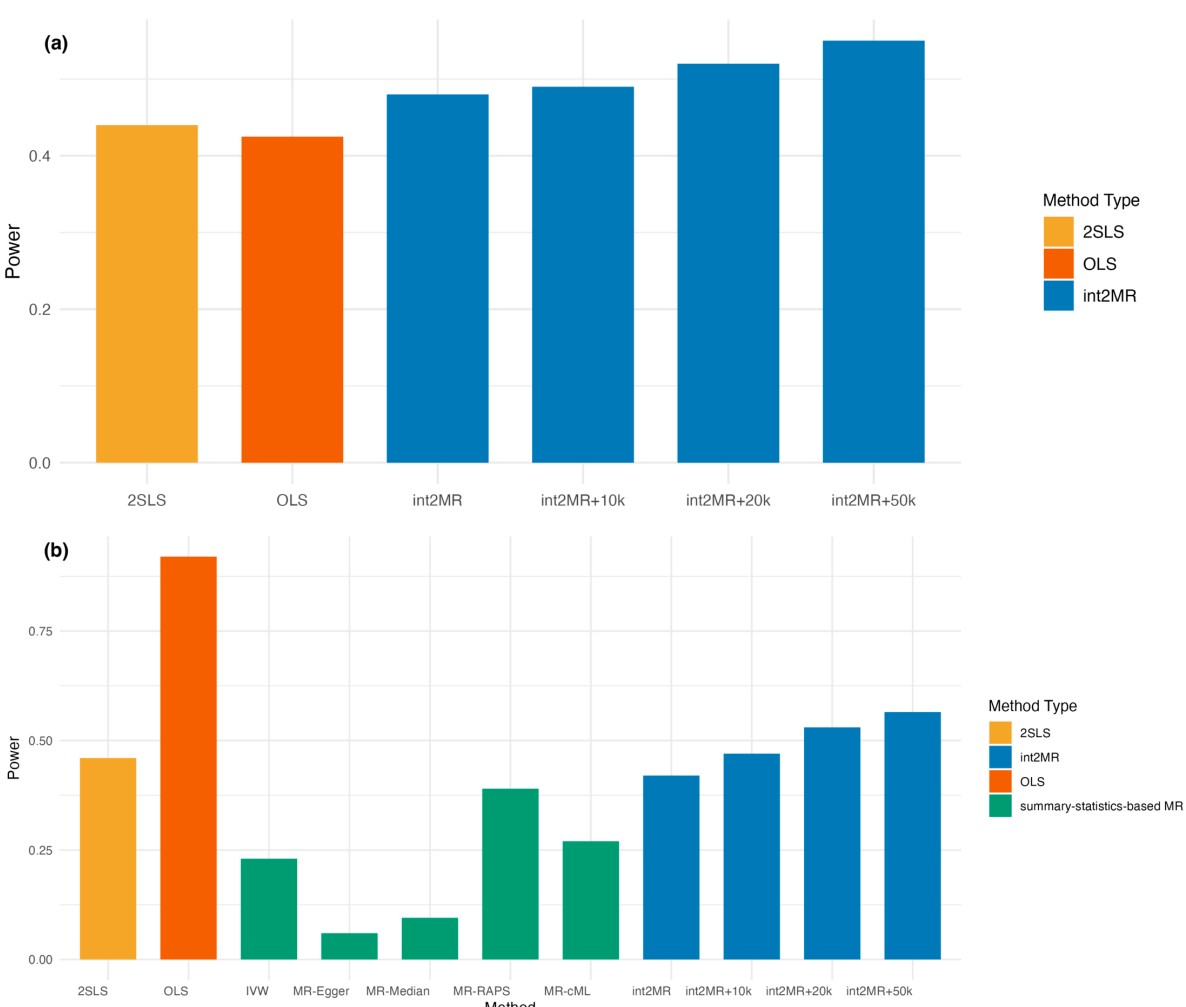

**Fig 2. Simulation results comparing the power of int2MR with competing methods.** (a) Power comparison for detecting interaction effects. The method int2MR$_{+N}$ integrates group-separated GWAS data with a group-combined GWAS dataset, where the subscript $N$ represents the sample size of the group-combined GWAS. Comparison methods are 2SLS and OLS-based interaction test, both requiring individual-level data. (b) Power comparison for detecting non-zero causal effects. We test for non-zero main or interaction effects for 2SLS, OLS-based interaction, and int2MR. For existing MR methods (colored green), we test for total effects.

(10k, 20k, and 50k) were integrated, int2MR showed a substantial power improvement. Fig 2B compares the power for detecting both main and interaction effects. For 2SLS, OLS, and int2MR, we tested non-zero main and interaction effects, while for other summary-based MR methods, we tested total effects. int2MR improved power by jointly testing for main and interaction effects.

In our int2MR method, integration of additional group-combined GWAS data substantially increases the sample size and thereby boosts the power, which is comparable or even higher than that of OLS. OLS relies solely on the available group-specific data (with a 2000-sample size of the reference group and a 1000-sample size of the comparison group). Without integrating the additional group-combined GWAS dataset, OLS has higher power than int2MR in detecting both the main effect and the shared effect. In contrast, int2MR leverages

the extra information provided by the group-combined data, which is not utilized by other individual-level MR methods.

In summary, the simulations showed that int2MR effectively controls type I error rates in the presence of pleiotropy and confounding. In addition, int2MR demonstrated strong power in detecting interaction effects and group-specific causal effects. The integration of group-combined and group-separated GWASs on outcomes further enhanced its power.

## Applications

### Data analysis: Identifying risk factors with sex-biased effects on ADHD

Attention-deficit/hyperactivity disorder (ADHD) is a sex-biased condition, with males significantly more likely to be diagnosed than females [51–53]. These observed sex differences suggest that certain risk factors may have varying causal effects depending on sex. Identifying and understanding these sex-specific effects and their mechanisms are crucial for improving diagnosis and developing targeted interventions for both sexes. In this analysis, we leverage existing GWAS summary statistics to systematically identify potential risk factors with sex-biased effects on ADHD.

We applied the proposed int2MR method to evaluate the interaction effects between risk exposures and sex groups on ADHD. The IV-to-exposure statistics were obtained from publicly available GWASs for 51 complex traits (see S1 Table) and diseases related to immunology, metabolism, gastrointestinal health, cardiovascular health, dermatological conditions, and brain function. The IV-to-outcome GWAS statistics on ADHD were obtained from the Psychiatric Genomics Consortium (PGC), including two sex-stratified ADHD GWAS datasets as well as a sex-combined dataset. The male-only GWAS included 32,102 individuals of European ancestry, while the female-only GWAS included 21,191 individuals of European ancestry [54]. Additionally, we used a larger sex-combined GWAS dataset consisting of 224,534 individuals, approximately 49.61% females [43]. We first applied int2MR using only sex-stratified ADHD GWAS statistics as the IV-to-outcome statistics. Additionally, we expanded the int2MR analysis by integrating sex-stratified GWAS statistics with a larger sex-combined GWAS to enhance the power for detecting interaction effects. For each exposure, SNPs significantly associated with the exposure ($p$-values $\leq 5 \times 10^{-8}$) were selected as IVs, followed by LD clumping with an $r^2$ threshold of 0.05. In this analysis, we apply LD clumping to select a set of relatively independent IVs without considering their LD structures. The analysis was restricted to 35 exposures with at least 20 IVs and at most 1000 IVs after IV selection.

At a false discovery rate (FDR) threshold of 0.1, we identified 15 traits with significant sex-biased effects on ADHD by integrating the large sex-combined GWAS. In comparison, int2MR using only sex-stratified GWASs identified 11 of these exposures. See Table 2 for the list of fifteen exposures with significant sex-interaction effects on ADHD. Overall, we observed more significant $p$-values for the exposures when integrating the sex-combined GWAS. The results demonstrate improved power in detecting interaction effects by integrating sex-combined GWAS summary statistics.

Fig 3 highlights several immune-related traits with sex-biased effects on ADHD, including white blood cell count, reticulocyte count, and granulocyte count, all of which showed larger effects (in absolute value) in males compared to females. These findings are consistent with previous reports on immune-related risk factors for ADHD [55–57], suggesting that immune system activity may play a more significant role in ADHD development among males. Prior studies have also shown that males are more prone to increased inflammatory responses, potentially driven by hormonal influences such as testosterone, which promotes a pro-inflammatory state [58]. Additionally, hypertension has been identified as a common

**Table 2. Results on fifteen risk exposures with significant sex-interaction effects on ADHD (FDR ≤ 10%) identified using int2MR.** The left panel shows the results based on integrating sex-stratified with sex-combined GWAS statistics. The right panel shows the results based on int2MR using sex-stratified GWAS statistics only.

| Exposure | #IVs | Sex-stratified GWAS only | | | + Sex-combined GWAS | | |
|---|---|---|---|---|---|---|---|
| | | $\widehat{\beta}_{int}$ | *p*-value | FDR adj. | $\widehat{\beta}_{int}$ | *p*-value | FDR adj. |
| High Light Scatter Reticulocyte Count | 676 | 0.117 | 0.0021 | 0.0611 | 0.125 | 0.0013 | 0.0312 |
| Reticulocyte Count | 681 | 0.089 | 0.0147 | 0.0644 | 0.100 | 0.0036 | 0.0312 |
| Sum of Neutrophil and Eosinophil Counts | 412 | 0.113 | 0.0146 | 0.0644 | 0.124 | 0.0044 | 0.0312 |
| White Blood Cell Count | 503 | 0.101 | 0.0042 | 0.0611 | 0.110 | 0.0037 | 0.0312 |
| Hayfever or Eczema | 353 | -0.371 | 0.0095 | 0.0611 | -0.412 | 0.0045 | 0.0312 |
| Self-Reported Hypertension | 407 | -0.412 | 0.0086 | 0.0611 | -0.415 | 0.0065 | 0.0359 |
| Self-Reported Psoriasis | 190 | 0.210 | 0.0105 | 0.0611 | 0.221 | 0.0072 | 0.0359 |
| Granulocyte Count | 409 | 0.117 | 0.0100 | 0.0611 | 0.125 | 0.0110 | 0.0479 |
| Myeloid White Cell Count | 418 | 0.096 | 0.0257 | 0.0960 | 0.102 | 0.0143 | 0.0557 |
| Neutrophil Count | 418 | 0.104 | 0.0343 | 0.1000 | 0.117 | 0.0198 | 0.0629 |
| Eosinophil Count | 601 | 0.078 | 0.0296 | 0.0960 | 0.080 | 0.0198 | 0.0629 |
| Sum of Basophil and Neutrophil Counts | 415 | 0.108 | 0.0302 | 0.0960 | 0.112 | 0.0231 | 0.0674 |
| Lymphocyte Count | 556 | 0.0783 | 0.0374 | 0.1010 | 0.0817 | 0.0275 | 0.0742 |
| Sum of Eosinophil and Basophil Counts | 560 | 0.0754 | 0.0601 | 0.1403 | -0.0800 | 0.0333 | 0.0787 |
| Inflammatory Bowel Disease | 155 | -0.0272 | 0.1997 | 0.3495 | -0.0389 | 0.0337 | 0.0787 |

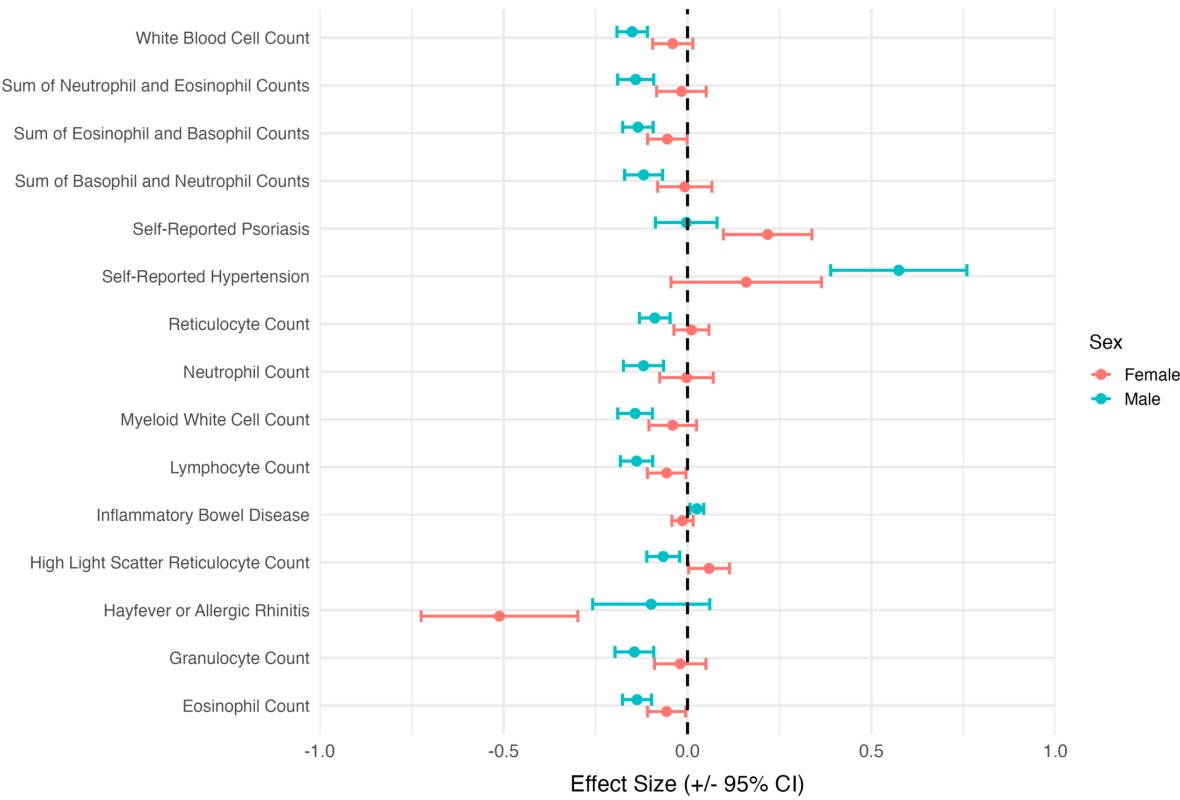

**Fig 3. A forest plot of causal effects for risk factors with significant sex-biased effects, stratified by sex (female vs. male).** Many of these risk factors are immune-related traits. Blue points represent male-specific effects, while red points represent female-specific effects.

comorbidity in adults with ADHD [59]. We observed consistent effect estimates across the two int2MR analyses, with or without the integration of sex-combined GWAS data. By leveraging existing GWAS summary statistics for risk exposure traits and outcome traits from multiple data sources and studies, int2MR enables an efficient evaluation of interaction effects across a wide range of exposures. This is particularly valuable for assessing interactions of risk exposures that are not measured in the studies of outcome traits, making int2MR a powerful and versatile tool for addressing research questions that would be infeasible to study using interaction analysis methods requiring individual-level data.

## Data analysis: Identifying age-group-specific risk factors for Alzheimer's disease in the oldest-old

AD and related dementias (ADRD) present a significant public health challenge, particularly as life expectancy continues to rise. Age is the primary risk factor for ADRD. A recent analysis of data from the Religious Orders Study and Rush Memory and Aging Project (ROSMAP) [45,60,61], involving 1,420 autopsied individuals, found that the probability of Alzheimer's dementia and cognitive impairment increases with age. Interestingly, a nonlinear relationship was observed for AD pathology, which peaks around age 95 and then slightly declines [62]. This pattern was evident in several AD pathology measures, including global AD pathology burden (gpath), amyloid, and PHF tau tangle density (tangles). In contrast, non-AD pathologies, except for TDP-43, continue to increase beyond age 95 in severity. Survival bias is unlikely, as the nonlinear trajectory with age is observed only for AD pathologies, while non-AD pathologies increase linearly with age. These findings highlight the need to understand the unique biology and mechanisms of neuropathologies in the oldest-old (death-age 95+) to develop effective prevention and treatment strategies for this rapidly growing age group.

To identify risk factors with age-group-specific effects on AD and pathologies in the oldest-old compared to the rest, we applied the int2MR method. We examined the same 51 complex traits and diseases as exposures that were used in our previous analysis on ADHD. We obtained the GWAS summary statistics for clinically diagnosed AD, three AD pathologies (gpath, amyloid, tangles), and three non-AD pathologies including TDP-43 (tdp_st4_binary)[63–67], hippocampal sclerosis (hspath_typ) [67–69], and cortical Lewy body (dlbany) [70–72], from the ROSMAP study. The ROSMAP GWAS data included 2,587 individuals, 408 of whom were 95 years or older at the time of death. We also obtained the GWAS summary statistics for only the oldest-old (N=408). For each exposure, SNPs with a $p$-value $\leq 5 \times 10^{-8}$ were selected as IVs, followed by LD clumping with an $r^2$ threshold of 0.05. The risk exposures and their age-group interaction effect estimates on three AD pathologies obtained in the int2MR analysis are presented in Table 3.

Fig 4 presents a heatmap showing the significance of age-group-interaction effects of selected exposures (FDR$\leq 0.05$) on AD pathologies (bottom panel) versus non-AD pathologies (top panel). The heatmap showed that several inflammation- and immune-related traits and diseases have significant age-group-interaction effects comparing the oldest-old to the others (95+ vs. 95-). Traits such as lymphocyte and eosinophil counts showed stronger associations with AD pathologies in younger individuals (below 95). However, these associations weaken or even reverse in the oldest-old, suggesting diminished immune responses in the oldest-old. In contrast, these risk factors do not have significant age-group interaction effects on non-AD pathologies. While immune-related exposures are strongly associated with non-AD pathologies, their associations are relatively consistent across age groups (95+ vs. 95-).

**Table 3. Risk exposures and their age-group interaction effect estimates on three AD pathologies obtained in the int2MR analysis.**

| Exposure | #IVs | $\widehat{\beta}_{int}$ | p-value | FDR adj. |
|---|---|---|---|---|
| **Amyloid Pathology** | | | | |
| Hair or Balding Pattern 4 | 710 | -0.5482 | $6.23 \times 10^{-5}$ | 0.00336 |
| Inflammatory Bowel Disease | 243 | -0.2078 | $2.73 \times 10^{-4}$ | 0.00736 |
| **Global AD Pathology Burden** | | | | |
| White Blood Cell Count | 833 | -0.2355 | $2.35 \times 10^{-7}$ | 0.00001 |
| Systemic Lupus Erythematosus | 251 | 0.0630 | $6.77 \times 10^{-6}$ | 0.00018 |
| Self-Reported Psoriasis | 315 | -0.6228 | $1.89 \times 10^{-5}$ | 0.00034 |
| Lymphocyte Count | 915 | -0.1795 | 0.00020 | 0.00177 |
| Myeloid White Cell Count | 686 | -0.2135 | 0.00018 | 0.00177 |
| Neutrophil Count | 660 | -0.2128 | 0.00014 | 0.00177 |
| Sum of Neutrophil and Eosinophil Counts | 656 | -0.2056 | 0.00083 | 0.00637 |
| Granulocyte Count | 651 | -0.2110 | 0.00182 | 0.01101 |
| Fluid intelligence score | 174 | 0.2522 | 0.00183 | 0.01101 |
| Sum of Basophil and Neutrophil Counts | 659 | -0.2138 | 0.00283 | 0.01389 |
| Sum of Eosinophil and Basophil Counts | 934 | -0.1353 | 0.00598 | 0.02690 |
| **Tangles Pathology** | | | | |
| Chronotype | 292 | 1.3546 | 0.00020 | 0.00528 |
| Self-Reported Psoriasis | 315 | -1.2458 | 0.00018 | 0.00528 |
| Crohn's Disease | 192 | 0.1629 | 0.00080 | 0.01432 |

The three AD pathologies analyzed are Amyloid Pathology (amyloid), Global AD Pathology Burden (gpath), and Tangles Pathology (tangles). Significant exposure-by-age-group interactions were identified at a 5% FDR threshold.

These findings underscore distinct pathological mechanisms in the oldest-old compared to the rest. The role of neuroinflammation in AD progression has been well-documented, with inflammation and immune responses driving the accumulation of AD pathology [73–75]. However, the reduced association between immune traits and AD in the oldest-old suggests a decline in immune response and neuroinflammatory activity, potentially explaining the plateau in AD pathology accumulation in this age group. This may be due to an age-related decline in the immune system's ability to mount an inflammatory response, along with the brain's compensatory mechanisms [76–81]. In contrast, non-AD pathologies continue to increase with age [62,82], which may result from a continuous and escalating inflammatory response that persists or intensifies with aging [83–88].

The underlying risk factors and molecular mechanisms driving these patterns in the oldest-old remain largely unexplored. Understanding the biology of AD and related pathologies is critical for developing targeted strategies for prevention, progression, and treatment. The proposed int2MR method demonstrates its advantage by leveraging age-group-specific GWAS statistics from ROSMAP, even with a limited sample size (N=408), and integrating them with publicly available GWAS data on diverse exposures. It enables the comprehensive evaluation of age-group interaction effects across a wide range of risk exposures, many of which are not directly measured in the ROSMAP dataset. Such evaluations would be infeasible with traditional interaction tests.

## Discussion

In this study, we introduced int2MR, an integrative MR method for detecting exposure-by-group interaction effects by leveraging GWAS summary statistics. int2MR combines group-separated and group-combined GWAS statistics on outcomes with GWAS statistics on exposures. The ability to use only summary statistics provides unparalleled flexibility, enabling

## Heatmap of Z Score

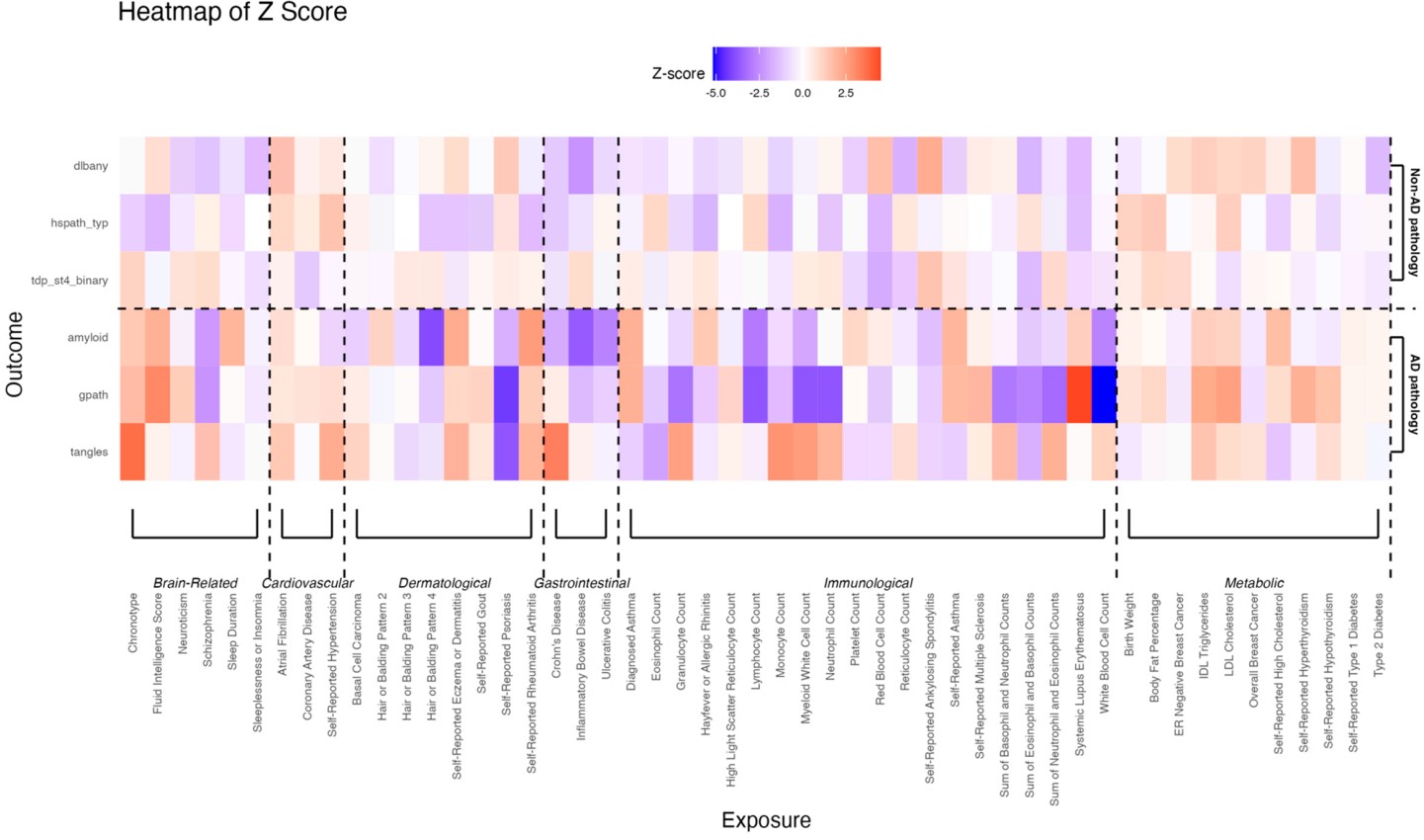

**Fig 4. A heatmap showing age-group interaction effects on AD pathologies (bottom panel) and non-AD pathologies (top panel), comparing the oldest-old to younger age groups (95+ vs. 95-).** Significant interaction effects were observed for several immune-related traits on AD pathologies. In contrast, these risk exposures showed fewer differences in effects comparing the two age groups, i.e., weaker age-group interactions for non-AD pathologies.

the evaluation of group-specific effects of risk factors on complex diseases when individual-level data are incomplete or unavailable. Our simulation studies demonstrated the high power and robustness of int2MR. The method consistently controlled type I error rates in various settings. In terms of power, int2MR showed reasonable performance when using group-separated GWAS summary statistics, comparable to analyses based on individual-level data. When integrating additional group-combined GWAS data on outcomes, int2MR had substantial power improvements.

We applied int2MR to identify exposure-by-sex interaction effects for ADHD, a sex-biased disorder. The analysis revealed multiple risk factors suggesting increased inflammatory responses in males. These findings are consistent with prior studies linking inflammation and immune dysregulation to ADHD. Importantly, the integration of sex-combined GWAS data improved the power to detect interaction effects, demonstrating the method's ability to leverage additional data to improve power. We further applied int2MR to identify age-group-specific risk factors for AD pathologies, with a focus on the individuals aged 95 and older (the oldest-old). This analysis identified multiple inflammation and immune-related risk factors, suggesting that chronic inflammation is reduced in the oldest-old. This may reflect an age-related decline in the immune system's ability to mount inflammatory responses, coupled

with compensatory mechanisms in the brain. In both analyses, int2MR integrated group-separated GWAS statistics on outcomes with GWAS statistics on exposures from various sources. This capability is a major advantage over existing methods that require individual-level data, allowing for the broad assessment of interactions and group-specific effects of risk factors for complex diseases.

Despite its strengths, int2MR has several challenges that present opportunities for future research. First, the current MR model assumes that IVs are not associated with unmeasured confounders, an assumption relaxed by some recent MR methods for total effects. In particular, if the correlated pleiotropic effect differs among the GWAS samples, the resulting pattern may confound the interaction effect ($\beta_{\text{int}}$) with the pleiotropic effect. In these cases, it is difficult to disentangle whether the observed deviations in the IV-to-outcome associations are due to a true interaction or to systematic pleiotropic biases. Future work could explore model extensions in this direction. Second, int2MR currently supports only binary group variables. Expanding the framework to interactions with continuous or multi-category variables would significantly broaden its utility. Our current work focuses on linear models. In many cases, especially when effect sizes are modest, the linear approximation provides a reasonable first-order approximation for binary outcomes. We plan to incorporate binary outcomes and nonlinear relationships in our future studies and refine the robustness and applicability of our methods under these conditions. Another opportunity is the application of int2MR in transcriptome-wide MR analyses, enabling the detection of genetically regulated genes and molecular risk factors with interaction effects across various cellular contexts. int2MR is a flexible, robust, and scalable tool for detecting exposure-by-group interaction effects. Its ability to integrate diverse GWAS summary statistics opens new avenues for understanding the complex interplay between risk factors and group-specific disease mechanisms.

## Supporting information

**S1 Text.** This document presents a comprehensive description of our methodological framework, including an extended discussion of the Bayesian hierarchical model and rigorous justification of all hyperparameter choices. It further details the simulation design and the procedures employed for generating summary statistics. S1 Text reports additional simulation results and expands upon the data-analysis findings presented in the main text.
(PDF)

**S1 Table.** List of the 51 genome-wide association study (GWAS) traits included in our analysis. For each trait, we report the phenotype name, total sample size, number of cases (where applicable), the contributing consortium or study, the citation for the primary GWAS publication (via PubMed), and the web portal used to access the full summary-statistic dataset.
(XLSX)

**S1 Checklist.** The STROBE-MR checklist of recommended items to address in reports of Mendelian randomization is included. This checklist is licensed under the Creative Commons Attribution 4.0 International License (CC BY 4.0; https://creativecommons.org/licenses/by/4.0/) and must be attributed to the STROBE Initiative. For more information, please see https://www.strobe-statement.org/.
(DOCX)

## Acknowledgments

The authors would like to thank Dr. Guimin Gao for valuable discussions.

## Author contributions

**Conceptualization:** Lin S. Chen.

**Formal analysis:** Ke Xu, Nathaniel Maydanchik.

**Methodology:** Ke Xu, Lin S. Chen.

**Resources:** Shinya Tasaki, David A. Bennett.

**Supervision:** Lin S. Chen.

**Writing – original draft:** Ke Xu, Lin S. Chen.

**Writing – review & editing:** Ke Xu, Nathaniel Maydanchik, Bowei Kang, Jianhai Chen, Qixiang Chen, Gongyao Xu, Shinya Tasaki, David A. Bennett, Lin S. Chen.

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
