## [Decision Letter · Decision Letter 0]

18 Mar 2025

PGENETICS-D-25-00150

Integrative Mendelian randomization for detecting exposure-by-group interactions using group-specific and combined summary statistics

PLOS Genetics

Dear Dr. Chen,

Thank you for submitting your manuscript to PLOS Genetics. After careful consideration, we feel that it has merit but does not fully meet PLOS Genetics's publication criteria as it currently stands. Therefore, we invite you to submit a revised version of the manuscript that addresses the points raised during the review process.

Please submit your revised manuscript within 60 days May 17 2025 11:59PM. If you will need more time than this to complete your revisions, please reply to this message or contact the journal office at plosgenetics@plos.org. Please include the following items when submitting your revised manuscript:

We look forward to receiving your revised manuscript.

Kind regards,

Xiaofeng Zhu

Section Editor

PLOS Genetics

Xiaofeng Zhu

Section Editor

PLOS Genetics

Aimée Dudley

Editor-in-Chief

PLOS Genetics

Anne Goriely

Editor-in-Chief

PLOS Genetics

**Journal Requirements:**

2) Your manuscript's sections are not in the correct order.  Please amend to the following order: Abstract, Author Summary, Introduction, Description of the Method, Verification and Comparison, Applications, Discussion, Acknowledgements, References, and Supplementary Information

4) We have noticed that you have uploaded Supporting Information files, but you have not included a complete list of legends. Please add a full list of legends for your Supporting Information files after the references list.

Potential Copyright Issues:

i) Figure 1. Please confirm whether you drew the images / clip-art within the figure panels by hand. If you did not draw the images, please provide (a) a link to the source of the images or icons and their license / terms of use; or (b) written permission from the copyright holder to publish the images or icons under our CC BY 4.0 license. Alternatively, you may replace the images with open source alternatives. See these open source resources you may use to replace images / clip-art:

2) State what role the funders took in the study. If the funders had no role in your study, please state: "The funders had no role in study design, data collection and analysis, decision to publish, or preparation of the manuscript.".

**Reviewers' comments:**

Reviewer's Responses to Questions

**Comments to the Authors:**

**Please note that one of the reviews is uploaded as an attachment.**

Reviewer #1: The authors propose a novel MR method to test for interaction effect of an exposure and a binary group on the outcome using group-stratified and group-combined GWAS summary statistic. The application on ADHD and AD revealed some interesting interaction effect with sex and age on the diseases. The idea is interesting and the paper is well structured and well written. I have several comments for improvement:

1. It might be better to move the model detail in the Supplementary S1.1 to the main text. This is insightful to motivate the main relationship of GWAS effects. There are several typos in the derivation, e.g. the equation after S2, $p$ should be *ρ*, and RHS is not conditioning on X.

2. Related to 1, could the authors provide more details on the assumptions and identifiability conditions of the model?

a) For example, what is the definition of valid/invalid IVs? What are the conditions such that *β* and *β_int_* are identifiable? The current model implicitly assume the group specific genetic effect on the outcome only due to the group specific effect of the exposure on the outcome (if no pleiotropy is present), is this correct? Group specific genetic effect on the exposure is not allowed?

b) How is the model distinguish $\Gamma_j = \beta \gamma_j + \alpha_j^*$, where $\alpha_j^* = \beta_{int}\gamma_j + \alpha_j$ (I assume this might also be related to the assumption of uncorrelated balanced pleiotropy)? An explicit and detailed description of model assumptions will be very helpful.

c) Based on the current model $\Gamma_j = (\beta + \beta_{int}\rho)\gamma_j) + \alpha_j$, is it possible to infer parameters of interest using only multiple combined outcome GWAS dataset with known *ρ*'s?

3. How is the power of testing \beta and \beta_{int} related to the sample sizes of different GWAS datasets? If this cannot be derived analytically, a more extensive simulation studies will be helpful.

4. For the simulation results, besides Type-I error and power, could the authors also show the averaged estimates, MSE, bias etc. (at least in the supplementary)?

5. In simulation, could the authors also consider scenarios with directional pleiotropy and/or correlated pleiotropy? It will be interesting to see how robust the model is in these scenarios.

6. In real data application, what are the results for shared effects $\beta$? Could the authors also present some results only based on combined GWAS dataset using existing MR methods, and discuss the comparison?

7. Line 234 mentioned 35 exposures are examined, but line 242 mentioned "the analysis was restricted to 51 exposures". And Fig4 show neither 35 nor 51 exposures. Which one is correct?

8. What is the result of clinically diagnosed AD?

9. The current heatmap only shows the difference in effect, could the authors also present a heatmap comparing the total effects in the two groups? This may give more information than only presenting the difference.

Reviewer #2: The authors propose int2MR, an integrative Mendelian randomization (MR) method that utilizes GWAS summary statistics for exposure traits and group-separated or combined GWAS statistics for outcome traits, all without requiring individual-level GWAS data. Overall, the manuscript is well-written, but I have a few suggestions for improvement:

In models (1-3), it is unclear how the authors handle (\gamma_j). Is it treated as a random variable with a prior, or as a fixed but unknown parameter?

The authors state that "int2MR improved power by jointly testing for main and interaction effects." This implies that the main effect should be non-zero. Two clarifications are needed: What if the main effect is zero? Additionally, can the method focus solely on testing the interaction effect?

In the real data analysis, how did the authors specify (\rho_k)? Is it known or unknown in practice?

Regarding sensitivity analysis, the authors used thresholds of (5 \times 10^{-8}) (ADHD) or (10^{-8}) (Alzheimer’s Disease) to select instrumental variables (IVs) and then performed LD clumping with an (r^2) threshold of 0.05, assuming all IVs are valid. Please conduct sensitivity analyses to ensure the robustness of the results.

Reviewer #3: The review is uploaded as an attachment.

**Have all data underlying the figures and results presented in the manuscript been provided?**

Reviewer #1: None

Reviewer #2: Yes

Reviewer #3: Yes

PLOS authors have the option to publish the peer review history of their article (what does this mean?). If published, this will include your full peer review and any attached files.

Reviewer #1: No

Reviewer #2: **Yes: **Can Yang

Reviewer #3: No

**Figure resubmission:**
---

## [Decision Letter · Decision Letter 1]

13 Jun 2025

PGENETICS-D-25-00150R1

Integrative Mendelian randomization for detecting exposure-by-group interactions using group-specific and combined summary statistics

PLOS Genetics

Dear Dr. Chen,

Thank you for submitting your manuscript to PLOS Genetics. The revised manuscript has been seen by the reviewers. Reviewer 2 is satisfied with this revision. However, reviewer 1 and 3 still have some concerns. Therefore, we invite you to submit a revised version of the manuscript that addresses the points raised during the review process.

Please submit your revised manuscript within 30 days Jul 13 2025 11:59PM. If you will need more time than this to complete your revisions, please reply to this message or contact the journal office at plosgenetics@plos.org. Please include the following items when submitting your revised manuscript:

We look forward to receiving your revised manuscript.

Kind regards,

Xiaofeng Zhu

Section Editor

PLOS Genetics

Xiaofeng Zhu

Section Editor

PLOS Genetics

Aimée Dudley

Editor-in-Chief

PLOS Genetics

Anne Goriely

Editor-in-Chief

PLOS Genetics

**Journal Requirements:**

Please ensure that the funders and grant numbers match between the Financial Disclosure field and the Funding Information tab in your submission form. Note that the funders must be provided in the same order in both places as well.

**Reviewers' comments:**

Reviewer's Responses to Questions

**Comments to the Authors:**

Reviewer #1: The authors have addressed most of my comments satisfactorily. I have two remaining comments:

1. Regarding Comment 1: The typo in the equation below line 71 appears to remain uncorrected and is the same as in the previous version.

2. Regarding Comment 6: I did not see the inclusion of results based on the combined GWAS dataset using existing MR methods. In addition, could the authors also perform traditional MR analyses using the sex-stratified GWAS data and compare those results with the findings presented in Figure S1?

Reviewer #2: Thank the authors' effort to address my concerns.

Reviewer #3: The authors have addressed some of my concerns, but several critical issues remain:

1. The response to comment is superficial. The data analysis used binary outcomes, and yet the proposed method is developed only for linear regression. This mismatch is unacceptable and must be rectified.

2. The rationale for omitting Zhu et al. (2024) for comparison is unpersuasive. By the same logic, OLS—also not designed for MR—should have been excluded.

3. The response states that “relatively independent” SNPs are required, yet the manuscript claims the method tolerates “moderate” correlation. Clarify what constitutes moderate correlation and soften the wording of generalization if necessary.

4. The explanation about power comparison in the response is strong, but the key points need to be integrated into the main text to prevent misinterpretation and ensure balanced representation.

**Have all data underlying the figures and results presented in the manuscript been provided?**

Reviewer #1: None

Reviewer #2: Yes

Reviewer #3: None

PLOS authors have the option to publish the peer review history of their article (what does this mean?). If published, this will include your full peer review and any attached files.

Reviewer #1: No

Reviewer #2: **Yes: **Can Yang

Reviewer #3: No

**Figure resubmission:**
---

## [Editor Report · Decision Letter 2]

25 Jul 2025

Dear Dr Chen,

We are pleased to inform you that your manuscript entitled "Integrative Mendelian randomization for detecting exposure-by-group interactions using group-specific and combined summary statistics" has been editorially accepted for publication in PLOS Genetics. Congratulations!

Yours sincerely,

Xiaofeng Zhu

Section Editor

PLOS Genetics

Xiaofeng Zhu

Section Editor

PLOS Genetics

Aimée Dudley

Editor-in-Chief

PLOS Genetics

Anne Goriely

Editor-in-Chief

PLOS Genetics

Comments from the reviewers (if applicable):

**Data Deposition**

http://datadryad.org/submit?journalID=pgenetics&manu=PGENETICS-D-25-00150R2

**Press Queries**

---

## [Editor Report · Acceptance letter]

PGENETICS-D-25-00150R2

Integrative Mendelian randomization for detecting exposure-by-group interactions using group-specific and combined summary statistics

Dear Dr Chen,

We are pleased to inform you that your manuscript entitled " 

Integrative Mendelian randomization for detecting exposure-by-group interactions using group-specific and combined summary statistics" has been formally accepted for publication in PLOS Genetics! Your manuscript is now with our production department and you will be notified of the publication date in due course.

With kind regards,

Benedek Toth

PLOS Genetics

On behalf of:
